# Coordination Sites for Sodium and Potassium Ions in Nucleophilic Adeninate Contact ion-Pairs: A Molecular-Wide and Electron Density-Based (MOWED) Perspective

**DOI:** 10.3390/molecules27186111

**Published:** 2022-09-19

**Authors:** Dominique M. S. Buyens, Lynne A. Pilcher, Ignacy Cukrowski

**Affiliations:** Department of Chemistry, University of Pretoria, Pretoria 0002, South Africa

**Keywords:** adeninate anion, nucleobases, alkali metals, ion-pairs, coordination modes, DMSO, computational chemistry, REP-FAMSEC, molecular-wide and electron density-based (MOWED) approach

## Abstract

The adeninate anion (Ade^−^) is a useful nucleophile used in the synthesis of many prodrugs (including those for HIV AIDS treatment). It exists as a contact ion-pair (CIP) with Na^+^ and K^+^ (M^+^) but the site of coordination is not obvious from spectroscopic data. Herein, a molecular-wide and electron density-based (MOWED) computational approach implemented in the implicit solvation model showed a strong preference for bidentate ion coordination at the N3 and N9 atoms. The N3N9-CIP has (i) the strongest inter-ionic interaction, by −30 kcal mol^−1^, with a significant (10–15%) covalent contribution, (ii) the most stabilized bonding framework for Ade^−^, and (iii) displays the largest ion-induced polarization of Ade^−^, rendering the N3 and N9 the most negative and, hence, most nucleophilic atoms. Alkylation of the adeninate anion at these two positions can therefore be readily explained when the metal coordinated complex is considered as the nucleophile. The addition of explicit DMSO solvent molecules did not change the trend in most nucleophilic N-atoms of Ade^−^ for the in-plane M-Ade complexes in M-Ade-(DMSO)_4_ molecular systems. MOWED-based studies of the strength and nature of interactions between DMSO solvent molecules and counter ions and Ade^−^ revealed an interesting and unexpected chemistry of intermolecular chemical bonding.

## 1. Introduction

Large anionic nucleophiles, generally depicted as Nu^−^ in chemical reaction schemes, are usually regarded as existing in their free form in solution; hence, the potential role of the counter ion is ignored. This has been the case for the adeninate anion, the deprotonated form of adenine, formed by the removal of the N9-H purine ring proton (pK_a_ = 9.8) in a polar aprotic solvent by a strong base, such as NaH, NaOH, or KOH [1,2,3,4]. The deprotonation of adenine is the first step in the synthesis of N9-alkylated adenine derivatives, such as the marketed prodrugs of tenofovir and adefovir used in the treatment of the human immunodeficiency virus (HIV) and hepatitis B virus (HBV) [5,6,7,8,9]. This step is followed by the direct alkylation of the adeninate anion [4,10,11,12,13,14,15,16,17]. The direct alkylation, conducted under different experimental conditions, leads to various mixtures of regio-isomers, having the N9-alkylated adenine reported as the main isomer in polar aprotic solvents [12,14,15,16,18,19,20]. The occurrence of mixtures of regio-isomers has sparked investigations to understand how the free anion governs alkylation at its four reactive nitrogen atoms, the N1, N3, N7, and N9 [15,18,19,21,22]. The common notion that the free adeninate anion carries an equal charge on all ring nitrogen atoms did not help to explain the regioselectivity of the alkylation reaction [18]. The choice of solvent has been shown to affect the regioselectivity, whereby increasing the ratio of water to DMSO shifts the major product from N9- to N3-alkylated adenine [15,19]. Furthermore, the alkylating reagent (mainly alkyl halides), leaving group, and the base counter ion (Li^+^, Na^+^, and K^+^) have been reported to show no effect on the regioselectivity [18,19,21,22]. Since the counter ions do not affect the alkylation ratio, it was proposed that they do not coordinate with the adeninate anion [19]. In general, the counter ion present in the solution is not considered to be coordinated with an anionic nucleophile, and studies into factors influencing alkylation/S_N_2 reactions usually only consider the free anionic nucleophile [23].

In contrast to the general perception, a recent NMR and UV spectroscopy study revealed that the adeninate anion does not exist as a free anion in DMSO, but rather as its contact ion-pair (CIP) with the base counter ions Na^+^ and K^+^ [24]. The interpretation of the NMR spectroscopic data proved to be challenging, and it was proposed that the base counter ions coordinate at either the N3 or N9 atom of the adeninate purine ring [24]. However, the experimental data could not provide insight into how the CIP’s reactivity towards an electrophile differs from that of the free anion.

As the formation of a CIP results in the rearrangement of the electron density throughout a molecular system, it becomes apparent that adeninate anions in CIPs must exhibit different nucleophilic properties compared to that of the free anion [25]. Hence, to understand the chemical behavior of the reaction system of a CIP, it is imperative to establish the coordination site for the metal cation. From the previous NMR experimental study on the CIP formation, the authors were unable to pinpoint and confirm where the counter ion is coordinated [24]. Therefore, combining experimental and computational studies should provide the most reliable models due to cross-validation between the two approaches. Hence, we decided to make use of the Interacting Quantum Atoms (IQA) electronic energy partitioning scheme [26] and the Reaction Energy Profile–Fragment Attributed Molecular System Energy Change (REP-FAMSEC) method [27]. This powerful combination has proved to be very useful in exploring and explaining molecules’ properties, reactivity, and reaction mechanisms on the fundamental atomic, functional group and molecular levels [27,28,29]. The IQA and REP-FAMSEC methods were employed here to explore the properties of Na- and K-Ade complexes using the recently reported Molecular-Wide and Electron Density-Based (MOWED) approach [29,30]. In the study reported here, essentially all intuitively possible Na^+^ and K^+^ adeninate complexes were investigated at the coupled cluster (CCSD) and density functional theory (DFT) levels to predict the preferred site and mode of cation coordination. We also report, for the first time, how the electronic environment of the adeninate anion changes the formation of CIPs in implicit and explicit (with four solvent DMSO molecules) solvation models. This phenomenon, i.e., the charge delocalization that leads to changes in the electronic environment throughout a molecular system, results in a new reactive environment that, when understood fully, should be invaluable in predicting chemical reactivity.

## 2. Results and Discussion

### 2.1. Data Obtained Using An Implicit Solvation Model

#### 2.1.1. Optimized Structures and Relative Energies

We began with the classical and generally accepted approach of computing the electronic energies of all possible CIPs of Na^+^ (Na-Ade) and K^+^ (K-Ade) adeninate complexes (M-Ade) to identify the CIP with the lowest energy. Optimized at the CCSD level, M-Ade structures of complexes considered in the implicit solvation model are shown in Figure 1; energy differences (relative to the lowest energy conformer) computed at both levels of theory (CCSD and DFT/B3LYP) are also included.

Two sets of CIPs were investigated, namely, (i) the in-plane complexes with the adeninate anion interacting with the cation through a specific N-atom, and (ii) the out-of-plane complexes with the adeninate anion interacting with the cation above the plane through a cation-π interaction or the amino group.

The energy of formation, $E_{f}$, was calculated using Equation (1),
(1)$$E_{f}=E_{\mathrm{M-Ade}}-\left(E_{{\mathrm{Ade}}^{-}}+E_{M^{+}}\right)$$
where $E_{\mathrm{M-Ade}}$ is the electronic energy of the CIP and $E_{{\mathrm{Ade}}^{-}}$ and $E_{M^{+}}$ (where M^+^ is Na^+^ or K^+^) are the electronic energy of the free adeninate anion and counter ion, respectively.

The largely negative $E_{f}$ CCSD-values obtained for each CIP (between −7 to −9 kcal mol^−1^ in the case of the in-plane complexes, see Figure 1) strongly suggests that the formation of CIPs in DMSO is spontaneous; notably, the highly comparable results of $E_{f}$ (between −7 and −10.5 kcal mol^−1^) were obtained at the DFT/B3LYP level. This is in support of the experimentally observed occurrence of CIPs in DMSO solution. The spontaneous formation of N3-, N9- and N3N9-CIPs of Na-Ade complexes was confirmed by the Gibbs free energy of formation ($G_{f}$), which was computed at the B3LYP level in the same fashion as $E_{f}$ in Equation (1); and, on average, a $G_{f}$ of −11.0 ± 0.3 kcal mol^−1^ was obtained.

The lowest energy CIPs have the counter ion coordinated (i) at the N3 and N9 atoms in a mono- or bidentate fashion for the Na-Ade complexes or (ii) only in bidentate coordination mode for the K-Ade complexes. There is little to no difference in electronic energy between the three lowest energy CIPs for the Na-Ade complexes, i.e., ΔE of 0 to 1 kcal mol^−1^. This supports the experimental prediction that the counter ions coordinate at these two N-atoms. The N7- and N1-CIP of the Na-Ade complexes are higher in energy than the N9-CIP by only ~1 and ~3 kcal mol^−1^, respectively. This energy difference is slightly larger for the K-Ade complexes where the N7- and N1-CIP are 2 to 4 kcal mol^−1^ higher in energy than the N3N9-CIP. These small ΔE values suggest, in contrast to the spectroscopic data, that at least four CIPs (N3, N9, N3N9, and N7) might be present at significant concentrations in DMSO solution [24]. It is then clear that classical computational protocol involving relative energy differences was not able to provide a decisive answer and failed to pinpoint the preferred complexation site.

#### 2.1.2. A Molecular-Wide Approach for Determining the Na^+^ and K^+^ Coordination Site(s) in the CIP

The molecular-wide and electron density-based concept of chemical bonding is based on the realization that any chemical event is driven by the interactions between entire molecules [30]. Naturally, not all atoms of the molecules will be playing a decisive and comparable role in driving a chemical change. Nonetheless, upon a chemical event, such as a conformational change or the formation of a new complex or adduct, there will be a resulting rearrangement of electron density and therefore, changes in charge distribution throughout the molecular system involving all atoms. This means that by considering such an event as only localized to the immediate reaction site, one might draw incorrect or invalid interpretations and conclusions [28,31].

REP-FAMSEC allows for the monitoring and quantifying of numerous energy terms, which can be uniquely defined for either the entire molecular system or its fragments. To explore and understand a chemical event, changes in these energy terms are monitored when moving from an initial reference (*ref*) state to a final (*fin*) state, such as moving from free ions (separated by 60 Å) to a CIP, Figure 1. The change in electron density, which, in essence, is a product of the chemical event, leads to a change in intra- and intermolecular interactions. Using REP-FAMSEC, which treats all CIPs on equal footing, we studied the interactions governing the formation of CIPs to identify the preferred site of ion coordination.

#### 2.1.3. The Inter-Ionic Interaction Energy of the CIPs of the Na- and K-Ade Complexes

The intermolecular interaction energy between the counter ion and the adeninate anion (Ade^−^), referred to as the inter-ionic interaction energy $E_{\mathrm{int}}^{M^{+},{\mathrm{Ade}}^{-}}$ (where M^+^ is Na^+^ or K^+^), can be classically seen as an attractive force that describes the strength of the CIPs formed. Mathematically, it is the summation of all unique IQA-defined diatomic interaction energies between the counter ion and all A atoms of the adeninate anion,
(2)$$E_{\mathrm{int}}^{M^{+},{\mathrm{Ade}}^{-}}=\underset{A\in {\mathrm{Ade}}^{-}}{\sum }E_{\mathrm{int}}^{M^{+},A}.$$

Any diatomic interaction energy is made of two major components, namely, a covalent/exchange-correlation XC-term ($V_{\mathrm{XC}}^{M^{+},A}$) and a classical electrostatic/Coulombic ($V_{\mathrm{cl}}^{M^{+},A}$) term,
(3)$$E_{\mathrm{int}}^{M^{+},{\mathrm{Ade}}^{-}}=\underset{A\in {\mathrm{Ade}}^{-}}{\sum }\left(V_{\mathrm{XC}}^{M^{+},A}+V_{\mathrm{cl}}^{M^{+},A}\right)=V_{\mathrm{XC}}^{M^{+},{\mathrm{Ade}}^{-}}+V_{\mathrm{cl}}^{M^{+},{\mathrm{Ade}}^{-}}.$$

The XC-term is related to the electron density shared (a covalent component of chemical bonding due to delocalization of electrons) between two ions, or atoms, involved in chemical bonding and the Coulombic term describes the electrostatic attraction between them.

The inter-ionic interaction energy, $E_{\mathrm{int}}^{M^{+},{\mathrm{Ade}}^{-}}$, and its covalent ($V_{\mathrm{XC}}^{M^{+},{\mathrm{Ade}}^{-}}$) and electrostatic ($V_{\mathrm{cl}}^{M^{+},{\mathrm{Ade}}^{-}}$) components computed for the formation of the CIPs are given in Table 1. Notably, the $E_{\mathrm{int}}^{M^{+},{\mathrm{Ade}}^{-}}$ values in Table 1 reveal that for both the Na- and K-Ade complexes at the CCSD level, the bidentate coordination at the N3 and N9 atoms is a staggering −30 kcal mol^−1^ more stabilizing than coordination at the N7. There is an even more significant stabilization of −40 kcal mol^−1^ found for the N3N9-CIPs relative to the N1 and N10 coordination sites for the Na- and K-Ade complexes and the π-CIP of the Na-Ade complex. These results demonstrate a significant preference for the N3 and N9 atoms as the site of coordination. The competition between unidentate and bidentate coordination at these two atoms amounts to about −10 kcal mol^−1^ in favor of the latter, with the Na^+^ forming a stronger interaction by −6.4 (CCSD) and −11.4 (DFT) kcal mol^−1^ than the K^+^. Importantly, the trends obtained at the CCSD level are qualitatively reproduced at the DFT level, showing that a comparative study can be performed on the latter with confidence. For the remaining sections, our focus is on CCSD results, whereas DFT data obtained in the implicit solvation model is placed in the Appendix A.

The data in Table 1 shows that, as one would predict, the interaction energy between M^+^ and Ade^−^ is dominated by the electrostatic attraction, the $V_{\mathrm{cl}}^{M^{+},{\mathrm{Ade}}^{-}}$ term. However, an unexpected and important covalent contribution of −10.8 ± 0.8 kcal mol^−1^ to the inter-ionic interaction energy term was found for all in-plane complexes of the Na-Ade complexes (see Figure 1). An even larger degree of covalent contribution was discovered for the K-Ade complexes for which we obtained the *V*_XC_ term of −15.8 ± 1.0 kcal mol^−1^. The most noteworthy contribution to the total inter-ionic interaction energy comes from the diatomic interactions between the counter ion and nitrogen atom to which the counter ions are coordinated (the Appendix A for the CCSD data and Appendix A for the DFT data) with additional and substantial contributions from the remaining nitrogen atoms. These stabilizing interactions outweigh the destabilizing contribution from the carbon-cation interactions.

#### 2.1.4. Influence of CIP Formation on the Intramolecular Interactions of the Adeninate Anion

Having understood the inter-ionic interactions, in terms of their strength and nature and how they vary with a coordination site, we turned our attention to the adeninate anion itself. This is because gaining further insight into the influence of the coordination sites on the intramolecular environment of Ade^−^, i.e., whether it (de)stabilized Ade^−^ itself, is of key interest and importance.

This was done by studying the change of the adeninate anion’s intramolecular interaction energy ($\Delta E_{\mathrm{int}}^{{\mathrm{Ade}}^{-}}$) using the free anion as the reference (*ref*) state ${\left(E_{\mathrm{int}}^{{\mathrm{Ade}}^{-}}\right)}_{ref}$ and the adeninate anion in its CIP form as a final (*fin*) state of the system, ${\left(E_{\mathrm{int}}^{{\mathrm{Ade}}^{-}}\right)}_{fin}$,
(4)$$\Delta E_{\mathrm{int}}^{{\mathrm{Ade}}^{-}}={\left(E_{\mathrm{int}}^{{\mathrm{Ade}}^{-}}\right)}_{fin}-{\left(E_{\mathrm{int}}^{{\mathrm{Ade}}^{-}}\right)}_{ref}.$$

As the total intramolecular interaction energy consists of contributions from all unique diatomic interactions between covalently bonded atoms, CB-interactions, (ECBintA,B) and the long-distance interactions (LD-interactions) between non-bonded atoms (ELDintA,B), $\Delta E_{\mathrm{int}}^{{\mathrm{Ade}}^{-}}$ can be expressed as
(5)ΔEintAde−=((∑A,B∈Ade−E CBintA,B)fin−(∑A,B∈Ade−E CBintA,B)ref)+((∑A,B∈Ade−E LDintA,B)fin−(∑A,B∈Ade−E LDintA,B)ref).

Equation (5) can be simply written as the sum of changes in the total interaction energy between covalently bonded atoms (ΔECBintAde−) and the long-distance interactions (ΔELDintAde−) between non-bonded atoms of the adeninate anion when moving from the *ref* to *fin* state,
(6)ΔEintAde−=ΔECBintAde−+ΔELDintAde−.

Relative to the free Ade^−^ anion, the changes in the total intramolecular interaction energy $\Delta E_{\mathrm{int}}^{{\mathrm{Ade}}^{-}}$, CB-interactions ΔECBintAde−, and LD-interactions ΔELDintAde− computed for Ade^−^ in M-Ade complexes are given in Table 2 (refer to Appendix A for the DFT data). The negative $\Delta E_{\mathrm{int}}^{{\mathrm{Ade}}^{-}}$ energy term reveals that the adeninate anion becomes significantly stabilized upon the formation of in-plane complexes due to the overall strengthening of intramolecular interactions. The degree of stabilization varies between −10.8 and −29.4 kcal mol^−1^ for the N1- and N3N9-CIP for the Na-Ade complexes and −4.6 and −19.9 kcal mol^−1^ for the N1- and N3N9-CIP for the K-Ade complexes. An opposite trend was discovered for the out-of-plane complexes where the adeninate anion became destabilized by +17.8/+2.8 kcal mol^−1^ for the N10-/π-CIP of the Na-Ade complexes and +13.5 kcal mol^−1^ for the N10-CIP for the K-Ade complex. Interestingly, the destabilization took place despite the highly attractive inter-ionic interactions shown in Table 1. Notably, data in Appendix A shows that the same general trends were recovered at the DFT level. The bidentate coordination of the counter ions at the N3 and N9 atoms stabilizes the adeninate anion by more than −8/−10 kcal mol^−1^ (CCSD/DFT) over the N9 site (slightly smaller values were obtained for the N3 coordination site). This once again points to these two atoms as highly favorable coordination sites for the CIP formation.

The stabilization of the adeninate anion for the in-plane complexes arises entirely from the strengthening of the CB-interactions of the adeninate anion, ΔECBintAde−, as the total intramolecular LD-interaction energy changed unfavorably with ΔELDintAde− > 0. Notably, the stronger the interactions between covalently bonded atoms, i.e., the ΔECBintAde− term becoming more negative, the more the LD-interactions weaken (Table 2). Conversely, the destabilization of the adeninate anion of the π- and N10-CIP is due to the weakening of the CB-interactions that override the strengthening of the LD-interactions experienced by the anion in these out-of-plane complexes.

To gain an insight into the origin of the strengthening/weakening of the CB-interactions (ΔECBintAde−), changes in their covalent *V*_XC_ and Coulombic *V*_cl_ components were investigated upon the formation of the CIPs. To achieve this, the change in the total intramolecular interaction energy for the adeninate anion was partitioned to relevant *V*_XC_ (exchange-correlation) and *V*_cl_ (classical Coulombic) terms of diatomic CB-interaction energies-Equation (7),
(7)ΔECBintAde−=((∑A,B∈Ade−V CBXCA,B)fin−(∑A,B∈Ade−V CBXCA,B)ref)+((∑A,B∈Ade−V CBclA,B)fin−(∑A,B∈Ade−V CBclA,B)ref),
that can also be expressed as a sum of changes in the total XC- and Coulombic terms as,
(8)ΔCBEintAde−=ΔVCBXCAde−+ΔVCBClAde−.

Changes in the XC- and Coulombic terms for the CB-interactions at the CCSD level are shown in Table 3 (refer to Appendix A for the DFT data). The slight weakening of the XC-term ΔVCBXCAde−, between +2.3 and +3.7 kcal mol^−^^1^ for N1- and N3N9-CIP of the Na-Ade complexes, and between 1.5 and 3.1 kcal mol^−^^1^ for the N1- and N3N9-CIP of the K-Ade complexes, represents an overall partial outflow of electron density from interatomic regions for all in-plane complexes. This, in turn, indicates a marginal overall decrease in the degree of the covalent character of intramolecular CB interactions on the formation of the in-plane complexes. This phenomenon can classically be interpreted as a decrease in the strength of the covalent bonds themselves [32]. By contrast, the electrostatic intramolecular interaction energy term changed in a highly stabilizing manner for all in-plane complexes as ΔVCBClAde− << 0 and, in absolute terms, is significantly larger than the change in the XC-term. This implies that an ion-induced polarization of the adeninate anion occurs when the counter ion coordinates, leading to increased (strengthened) electrostatic components of diatomic interactions between the covalently bonded atoms.

The Na^+^ ion-induced polarization, when measured by the ΔVCBClAde− term, is −42.5, −20.5, and −13.6 kcal mol^−1^ for the N3N9-, N7-, and N1-CIP of the Na-Ade complexes, respectively. The K-Ade complexes showed the same trend but to a lesser extent, i.e., −28.3, −9.3, and −5.1 kcal mol^−1^ for N3N9-, N7-, and N1-CIP, respectively. This shows that the ion-induced polarization is about twice as strong for the Na-Ade complexes as the K-Ade complexes. This is due to the Na^+^ ion perturbing the electronic environment to a larger extent than K^+^ as a result of the larger charge-to-radius ratio of Na^+^. In other words, the charge density of Na^+^ is significantly larger, rendering it more polarizing when it approaches another molecule/ion.

#### 2.1.5. Change in Net Atomic Charges upon CIP Formation

The ion-induced polarization of the adeninate anion may hold the key to predicting new reactive sites based on the rearrangement of electron density. From this information, new emerging properties of the CIPs themselves can be proposed. The intramolecular polarization can be explored using the net atomic charges, *Q*(A), of all atoms in the *fin* and *ref* states. This is because the change in net atomic charge, Δ*Q*(A), is the result of the rearrangement of the electron density of the adeninate anion upon CIP formation,
(9)$$\Delta Q\left(A\right)=Q{\left(A\right)}_{fin}-Q{\left(A\right)}_{ref},$$
where *Q*(A)*_ref_* and *Q*(A)*_fin_* are the net atomic charge of atom A of the adeninate anion prior (*ref* state) and after (*fin* state) CIP formation. The outflow or inflow of electron density upon complexation of the metal ion is indicated by a positive or negative Δ*Q* value, respectively.

The total net charge of the adeninate anion *Q*(Ade^−^) is obtained by the summation of the net atomic charges of all its atoms. The change, Δ*Q*(Ade^−^), of a net molecular (ionic) charge, Equation (10), represents either the resultant outflow from or inflow of density into the adeninate anion upon the formation of the CIP,
(10)$$\Delta Q\left({\mathrm{Ade}}^{-}\right)=\underset{A\in {\mathrm{Ade}}^{-}}{\sum }Q{\left(A\right)}_{fin}-\underset{A\in {\mathrm{Ade}}^{-}}{\sum }Q{\left(A\right)}_{ref}.$$

The change in the net atomic charge of the Na^+^ and K^+^ counter ion, Δ*Q*(M^+^), is calculated likewise, i.e., by subtracting the net atomic charge computed for the *ref Q*(M^+^)*_ref_* state from that obtained for the *fin* state *Q*(M^+^)*_fin_*,
(11)$$\;\Delta Q\left(M^{+}\right)=Q{\left(M^{+}\right)}_{fin}-Q{\left(M^{+}\right)}_{ref}$$

The full set of the CCSD data related to net charges is given in Table 4. Firstly, it is to be noted that the charges on the endo-purine ring nitrogen atoms of the free adeninate anion, before CIP formation, are comparable, i.e., −1.276 *e*, −1.266 *e*, −1.248 *e*, and −1.234 *e* for the N3, N1, N9, and N7 atoms, respectively. This trend is replicated in the DFT data (Appendix A). Hence, from a net atomic charge perspective, the Na^+^ and K^+^ counter ions can potentially coordinate to any of the four N-atoms of the endo-purine ring and, notably, M^+^ would be preferentially coordinated to N1- over the N9-atom due to a stronger electrostatic attraction between the oppositely charged atoms. This is in total contradiction to experimental data and computational MOWED-based data (included in Table 1) pointing to the preference of N9- over the N1-coordination site [24]. This is an excellent example showing that focusing on a single atom or atom-pair might lead to incorrect conclusions. The data in Table 4 reveals that ion-induced polarization is always the most significant at the coordination site. In particular, as indicated by the negative Δ*Q* value, the largest gain in electron density by N-atoms is observed for the bidentate N3N9-CIP coordination mode, i.e., −0.021 and −0.026 *e* for the N3 and N9 atoms, respectively, for the Na-Ade complex. Furthermore, on the formation of the mono-dentate complexes, the N3-CIP (−0.026 *e*) and N9-CIP (−0.022 *e*) show a larger inflow of electron density to N-atoms than the N7-CIP (−0.020 *e*) and N1-CIP (−0.017 *e*). In alignment with atomic charge density being larger in the case of Na^+^, the gain in electron density at the coordinated nitrogen atoms of the Na-Ade complexes is 2–3 times greater than that of the K-Ade complexes, demonstrating that the K^+^ ion perturbs the electron density to a much smaller extent.

From Table 4 it is also evident that the charge difference between neighboring atoms of Ade^−^ increased upon the formation of CIPs. This explains why the Coulombic term of CB-interactions strengthened to a significant extent (Table 3).

The overall positive change in the net molecular charge of the adeninate anion upon metal ion coordination, i.e., Δ*Q*(Ade^−^) > 0, shows an outflow of electron density to the counter ion. The charge transfer from the anion to the counter ions also explains a relatively large covalent contribution (the XC-term) of about 10–15% to the total inter-ionic interaction energy (Table 1). Notably, the outflow of electron density from the adeninate anion is similar between Na- and K-Ade complexes even though the ion-induced polarization is much smaller for K-Ade complexes. For both the Na^+^ and K^+^ complexes, the charge transfer is greatest for the N3N9-CIP, i.e., Δ*Q*(Ade^−^) = 0.039 *e* for the Na-Ade complex and 0.041 *e* for the K-Ade complex, and this correlates well with the strongest inter-ionic interaction shown in Table 1.

The identification of the most likely sites for ion coordination, along with ion-induced charge polarization of the entire Ade^−^, indicates that the ‘bottom’ part of Ade^−^ (containing N3- and N9-atoms) is the most polarizable, with these two atoms gaining the most negative charge. This strongly suggests that these coordination sites might be the most nucleophilic for alkylation with an incoming electrophile. In support of this, experimental data shows that the alkylation of the adeninate anion in DMSO solution leads to N9- (major) and N3- (minor) alkylated adenine products, with little or no formation of N7-alkylated adenine [15]. Finally, the consistent picture obtained from the computational modeling of the Na^+^ and K^+^ ions coordinating at the N3 and N9 atoms explains the same alkylation pattern reported when either counter ion is present.

### 2.2. Data Obtained Using An Explicit Solvation Model

The above in-depth computational investigations (using an implicit solvation model) agree very well with the experimental spectroscopic observation of Na- and K-Ade complexes in DMSO solution; it supports the suggestion that the counter ions coordinate at the N3 and N9 nitrogen atoms and points at these two atoms as the most likely nucleophilic sites [24]. While this provides a consistent picture, it is also highly desirable to consider the potential impact of explicit DMSO solvent molecules, which are known to form hydrogen bonds with the amino group of the adeninate anion [33]. Thus, the question arose as to whether or not the inclusion of explicit solvent molecules in the computational modeling of the Na- and K-Ade complexes will (i) change the nature and strength of inter-ionic M^+^⋯Ade^−^ interactions, (ii) influence the preferred site for counter ion coordination to Ade^−^, or (iii) predict the dissociation of some (or all) CIPs altogether.

As explained in the Computational Methods Section, we restricted our investigation to four DMSO molecules. Our focus here is on the properties of the M-Ade-(DMSO)_4_ molecular systems made of the in-plane Na- and K-Ade complexes as they featured prominently in the implicit solvation model. The π-CIP was not recovered using the explicit solvent molecules and the N10-CIP remained highly unfavorable with explicit DMSO molecules and will not be discussed further.

#### 2.2.1. Relative Energies of M-Ade-(DMSO)_4_ Systems and M-Ade Complexes in the Systems

The geometries for the in-plane Na-Ade-(DMSO)_4_ and K-Ade-(DMSO)_4_ optimized systems are shown in Figure 2 (see Appendix A for the out-of-plane N10-CIP). Notably, even though the M-Ade-(DMSO)_4_ systems were optimized without any constraints, two DMSO molecules (numbered DMSO-1 and DMSO-2) are essentially harbored at the adeninate –NH_2_ functional group irrespective of the site of metal ion coordination. This is due to the classical N–H⋯O hydrogen bond formed between the –NH_2_ group and O-atoms of the DMSO-1 and DMSO-2 molecules. The remaining two solvent molecules, numbered DMSO-3 and -4, are always interacting with M^+^ when it is coordinated to either N3 or N9. Most importantly, however, it is seen in Figure 2 that all Na-and K-Ade in-plane complexes are preserved in the M-Ade-(DMSO)_4_ systems. The energy of the M-Ade complexes within the M-Ade-(DMSO)_4_ systems was obtained by removing the explicit DMSO molecules and performing a single-point energy calculation on the M-Ade complex. The trend in relative stabilities of the in-plane M-Ade complexes is perfectly reproduced from the implicit solvation model and, focusing on sodium complexes, the N3N9-CIP is the most stable, whereas N1-CIP is the least stable. The later complex is higher in energy by about 3.3 kcal/mol (it was 3.2 kcal/mol in the implicit solvation model studies; see the Δ*E*_Na-Ade_ values in Figure 2). The N3- and N9-CIP are not significantly different in energy compared to the lowest energy N3N9-CIP, i.e., Δ*E*_Na-Ade_ of 0.4 and 1 kcal mol^−^^1^, respectively, as was observed for the implicit solvation model.

The energies of formation (*E*_f_ in Figure 2) for the in-plane complexes in the presence of four DMSO molecules were computed as the difference between the energy of the M-Ade-(DMSO)_4_ system and energies of reactants using Equation (12),
(12)$$E_{f}=E_{\mathrm{M-Ade-}{\left(\mathrm{DMSO}\right)}_{4}}-\left(E_{{\mathrm{Ade}}^{-}}+E_{M^{+}}+4E_{\mathrm{DMSO}}\right).$$

The formation of the M-Ade-(DMSO)_4_ systems is spontaneous as indicated by the large negative *E*_f_ values in all cases (in absolute terms *E*_f_ > 40 kcal mol^−1^). The free energy for the formation *G*_f_ of the M-Ade complex within the M-Ade-(DMSO)_4_ systems is also negative; on average, a $G_{f}$ of −3.8 ± 0.6 kcal mol^−1^ was obtained for the N3N9-, N3-, and N9-CIP of the Na-Ade-(DMSO)_4_ system, showing that the CIPs form spontaneously in the presence of explicit DMSO molecules.

In comparison to the *E*_f_ values from Figure 1, i.e., for complexes studied under the implicit solvation model conditions, a contribution of around −30 kcal mol^−1^ arises from the interactions of the Na- and K-Ade CIPs with the DMSO solvent molecules. It is then clear that the explicit DMSO molecules stabilize all the systems sizably but they do not have a significant impact on the formation and relative stability of the M-Ade solvated by DMSO complexes. Moreover, we note that the effect of the DMSO solvent molecules on the overall energy of the system is highly dependent on their placement; see the Δ*E*_system_ values in Figure 2 computed relative to the lowest energy M-Ade-(DMSO)_4_ system. The N7-CIP for the Na-Ade complexes and N7- and N1-CIPs for the K-Ade complexes appear to have the most favorable (energy-stabilizing) organization of DMSO solvent molecules with the adeninate anion and counter ion.

#### 2.2.2. Effect of Explicit Solvent Model on the Inter-Ionic Interactions 

The inter-ionic interaction energies $E_{\mathrm{int}}^{M^{+},{\mathrm{Ade}}^{-}}$ between M^+^ and Ade^−^ in the M-Ade-(DMSO)_4_ molecular systems were calculated using Equation (2). The data obtained, together with the covalent ($V_{\mathrm{XC}}^{M^{+},{\mathrm{Ade}}^{-}}$) and Coulombic ($V_{\mathrm{cl}}^{M^{+},{\mathrm{Ade}}^{-}}$) components of these interactions, are given in Table 5. Notably, the inter-ionic interactions between M^+^ and Ade^−^ in the M-Ade-(DMSO)_4_ systems weakened by about 10 kcal mol^−1^. For example, the $E_{\mathrm{int}}^{M^{+},{\mathrm{Ade}}^{-}}$ term obtained for the N3N9-CIP of the Na-Ade complex changed from −133.6 kcal mol^−1^, when the implicit solvation model was used (Table 1), to −124.5 kcal mol^−1^ in the Na-Ade-(DMSO)_4_ molecular system.

By comparing the data in Table 1 and Table 5, it is clear that the decrease in the strength of inter-ionic interactions stems from the weakening in both the covalent and Coulombic components. As an example, the covalent, $V_{\mathrm{XC}}^{{\mathrm{Na}}^{+},{\mathrm{Ade}}^{-}},$ and electrostatic, $V_{\mathrm{cl}}^{{\mathrm{Na}}^{+},{\mathrm{Ade}}^{-}},$ terms dropped at the DFT level from −18.6 and −115.0 (implicit solvation model, Table 1) to −14.5 and −110.0 kcal mol^−1^, respectively, in the presence of explicit DMSO molecules (Table 5). Importantly, however, trends discovered in the implicit solvation model are also valid in the presence of explicit DMSO molecules. To this effect, the bidentate coordination at the N3 and N9 atoms leads to the N3N9-CIP that is characterized by the strongest inter-ionic interactions. Furthermore, this interaction is still −30 and −40 kcal mol^−1^ more stabilizing than coordination at the N7 and N1 atoms, and −10 kcal mol^−1^ more stabilizing than unidentate coordination at either of the N3 and N9 atoms.

The agreement between the implicit and explicit DMSO solvation models eliminates any doubt that the inclusion of explicit solvent molecules might change the preferred coordination site of the counter ion. On the other hand, when using explicit solvent molecules that represent a more realistic reaction environment, a greater understanding of the molecular-wide chemical event involving all molecules in the molecular system can be gained. To this effect, and for the first time, in the following sections, we have explored the nature and strength of intermolecular interactions between the DMSO molecules and the counter ion as well as the adeninate anion.

#### 2.2.3. Strength and Nature of Interactions between DMSO Solvent Molecule and the Na^+^ and K^+^ Counter Ions in M-Ade-(DMSO)_4_ Molecular Systems

Equation (2), originally used to compute interaction energies between M^+^ and Ade^−^, was modified to involve a DMSO solvent molecule, $E_{\mathrm{int}}^{M^{+},\mathrm{DMSO}}$,
(13)$$E_{\mathrm{int}}^{M^{+},\mathrm{DMSO}}=\underset{A\in \mathrm{DMSO}}{\sum }E_{\mathrm{int}}^{M^{+},A},$$
and Equation (3) was expressed in such a way as to allow computing of the exchange-correlation ($V_{\mathrm{XC}}^{M^{+},\mathrm{DMSO}}$) and Coulombic ($V_{\mathrm{cl}}^{M^{+},\mathrm{DMSO}}$) components of the total interaction energy between M^+^ and a DMSO molecule,
(14)$$E_{\mathrm{int}}^{M^{+},\mathrm{DMSO}}=\underset{A\in \mathrm{DMSO}}{\sum }\left(V_{\mathrm{XC}}^{M^{+},A}+V_{\mathrm{cl}}^{M^{+},A}\right)=V_{\mathrm{XC}}^{M^{+},\mathrm{DMSO}}+V_{\mathrm{cl}}^{M^{+},\mathrm{DMSO}}.$$

The results for DMSO-3 and DMSO-4 are included in Table 6 and data obtained for DMSO-1 and DMSO-2 molecules are given in Appendix A. Considering the N3-, N9-, and N3N9-CIPs, the DMSO-3 and DMSO-4 molecules are always near the counter ion M^+^ if it is coordinated with the N3 and N9 atoms.

However, when a counter ion is placed at the N1 and N7 donor atoms, a remarkable movement of these two DMSO molecules, DMSO-3 and DMSO-4, occurs from where they were placed near the N3 and N9 atoms to near the site of metal coordination, N1 or N7. An inspection of diatomic interaction energies between all atoms of the M-Ade-(DMSO)_4_ systems (Appendix A) revealed that this movement of solvent molecules is due to the high affinity between the negatively charged DMSO’s O-atom and the positively charged counter ion. The interactions between Na^+^ and a DMSO’s O-atom are the strongest among all unique atom-pairs of the Na-Ade-(DMSO)_4_ system, i.e., an average $E_{\mathrm{int}}^{{\mathrm{Na}}^{+},O}$ of −168.7 kcal mol^−1^ was found between O37 of DMSO-3 and O47 of DMSO-4 in the case of N9-CIP. Moreover, considering the N1-CIP, the counter ion Na^+^ is involved in three interactions of nearly the same strength with O17, O37, and O47 of DMSO-1, DMSO-3, and DMSO-4, respectively.

The data in Table 6 shows a very small variation in the three interaction energy terms computed for the five Na-Ade-(DMSO)_4_ molecular systems where a DMSO molecule is directly involved in the interaction with a counter ion. Considering DMSO-3, on average for all CIPs, we obtained $E_{\mathrm{int}}^{{\mathrm{Na}}^{+},\mathrm{DMSO}-3}$ of −44.9 ± 1.5 kcal mol^−1^. This strongly suggests that the total interaction energy between a DMSO molecule and the Na^+^ counter ion approaches a constant value of about −45 kcal mol^−1^. A remarkable constancy is also observed for the interaction energy components as we obtained −11.2 ± 0.5 and −33.7 ± 1.3 kcal mol^−1^ for the $V_{\mathrm{XC}}^{{\mathrm{Na}}^{+},\mathrm{DMSO}-3}$ and $V_{\mathrm{cl}}^{{\mathrm{Na}}^{+},\mathrm{DMSO}-3}$ components, respectively. This, in turn, shows that the nature of the Na^+^⋯DMSO interaction remains unchanged regardless of the coordination site considered. However, the impact of an immediate environment can be seen in the interaction between DMSO-4 and the N7-CIP where a much weaker interaction is observed between this DMSO molecule and a counter ion due to the significant distance between them.

Quite surprisingly, we discovered that the covalent $V_{\mathrm{XC}}^{{\mathrm{Na}}^{+},\mathrm{DMSO}-3}$ contribution amounts to 25% of the total interaction energy. This implies that a significant amount of electron density is shared between a DMSO solvent molecule and its immediate counter ion neighbor, yet it does not largely influence the inter-ionic $E_{\mathrm{int}}^{{\mathrm{Na}}^{+},\mathrm{Ade}-}$ interaction. The nature of the interaction appears to depend on the counter ion. Focusing on the data in Table 6 obtained for DMSO-3 and K^+^, the interaction is dominated by the electrostatic component (as found for the Na^+^⋯DMSO interactions), but the degree of covalency reached a staggering 33% for the N3N9-, N7-, and N1-CIPs. As the cations are electron-deficient, it is reasonable to assume that the DMSO molecule shares its density with its neighbor, either Na^+^ or K^+^.

#### 2.2.4. Interactions of DMSO Solvent Molecules with the Adeninate Anion in M-Ade-(DMSO)_4_ Systems

By summing up the intermolecular diatomic interaction energies computed for all unique atom-pairs made of atoms A of DMSO and B of Ade^−^, one obtains the total intermolecular interaction energy, $E_{\mathrm{int}}^{{\mathrm{Ade}}^{-},\mathrm{DMSO}},$ between them,
(15)$$E_{\mathrm{int}}^{{\mathrm{Ade}}^{-},\mathrm{DMSO}}=\underset{A\in \mathrm{DMSO}}{\sum }\underset{B\in {\mathrm{Ade}}^{-}}{\sum }E_{\mathrm{int}}^{A,B}.$$

A similar protocol applies to the two components of the total interaction energy, $V_{\mathrm{XC}}^{{\mathrm{Ade}}^{-},\mathrm{DMSO}}$ and $V_{\mathrm{cl}}^{{\mathrm{Ade}}^{-},\mathrm{DMSO}}$,
(16)$$E_{\mathrm{int}}^{{\mathrm{Ade}}^{-},\mathrm{DMSO}}=\underset{A\in \mathrm{DMSO}}{\sum }\underset{B\in {\mathrm{Ade}}^{-}}{\sum }\left(V_{\mathrm{XC}}^{A,B}\right)+\underset{A\in \mathrm{DMSO}}{\sum }\underset{B\in {\mathrm{Ade}}^{-}}{\sum }\left(V_{\mathrm{cl}}^{A,B}\right)=V_{\mathrm{XC}}^{{\mathrm{Ade}}^{-},\mathrm{DMSO}}+V_{\mathrm{cl}}^{{\mathrm{Ade}}^{-},\mathrm{DMSO}}$$

Our analyses here focus on the DMSO-1 and DMSO-2 molecules as these two solvent molecules interact directly with the amino group of the adeninate anion through the classical hydrogen bonding of all Na- and K-Ade complexes. The results obtained, i.e., the $E_{\mathrm{int}}^{{\mathrm{Ade}}^{-},\mathrm{DMSO}}$ interaction energy and its components computed for the M-Ade-(DMSO)_4_ systems involving DMSO-1 and DMSO-2, are included in Table 7 and those obtained for DMSO-3 and DMSO-4 are given in Appendix A.

Considering the DMSO molecules near the amino group, the magnitude and the stabilizing nature of their interactions with the adeninate anion are highly comparable with those obtained for M^+^⋯DMSO interactions shown in Table 6. However, and in contrast to the interactions between the counter ion and either Ade^−^ or DMSO molecules, discussed above, the exchange-correlation $V_{\mathrm{XC}}^{{\mathrm{Ade}}^{-},\mathrm{DMSO}}$ term hugely dominates the interaction between the DMSO molecules and adeninate anion. This covalent component averages about 75% of the total interaction energy and approaches nearly 79% in the case of DMSO-1 interacting with Ade^−^ in the N3N9-CIP of the Na-Ade-(DMSO)_4_ molecular system. This is three-times larger than we found for the interactions between the counter ion and a DMSO molecule. In addition, the stabilizing contribution from the XC-term of about −30 kcal mol^−1^ is over twice as much as that found for the inter-ionic $V_{\mathrm{XC}}^{{\mathrm{Na}}^{+},{\mathrm{Ade}}^{-}}$ interaction (see Table 1). Notably, the classical term is relatively small and attractive; from this follows that the main ‘glue’ keeping DMSO molecules close to the –NH_2_ functional group of Ade^−^ is the electron density shared between them. As far as we could establish, there was no prior report of this nature and significance before.

## 3. Materials and Methods

### Computational Details

Na-Ade and K-Ade complexes were modeled by varying the coordination sites of the counter ions, Na^+^ and K^+^. Energy optimizations of the free adeninate anion and its Na- and K-Ade complexes were performed in Gaussian 09 rev. E.01 [34] at the DFT/B3LYP/6-311++g(d,p) level with Grimm’s GD3 empirical dispersion correction and in Gaussian 16 rev. B.01 [35] at the CCSD/6-311++g(d,p) level of theory. To verify that the structures obtained at the DFT/B3LYP level of theory are the true minima, frequency calculations were performed to ensure that no imaginary/negative frequency was present. All computations were performed in solvent (DMSO) using the Polarizable Continuum Model (PCM). Considering calculations involving explicit DMSO solvent molecules, the CIPs involving N10 of adeninate anion were excluded; the reasoning for such is explained in the sections that follow. To treat the remaining four N-atoms on equal footing, i.e., N1, N3, N7, and N9, and to minimize the computational cost, we decided to make use of four explicit DMSO solvent molecules. They were placed such that two of the DMSO molecules interacted with the –NH_2_ functional group and the other two DMSO molecules were at the ‘bottom’ of Ade^−^ being near N3 and N9. As the inclusion of solvent molecules increased the number of atoms by forty, we optimized the M-Ade-(DMSO)_4_ systems without any constraints only at the DFT/B3LYP/6-311++g(d,p)/GD3/PCM level of theory. IQA calculations were done using AIMAll (Version 19.10.12) [36]. Software developed in-house was used to run the REP-FAMSEC calculations [27].

## 4. Conclusions

In the earlier studies of the regioselectivity of the adeninate anion (Ade^−^), the nucleophile used in the synthesis of important prodrugs such as tenofovir and adefovir, the base counter ion was ignored or suggested not to be coordinated to the adeninate anion. However, it has been shown that Ade^−^ forms contact ion-pairs (CIPs) with Na^+^ and K^+^ (M) in DMSO solution [24]. It was proposed, from the spectroscopic study, that the counter ion coordinates at the N3- and N9-atoms of Ade^−^ but it was impossible to pinpoint and confirm where the counter ion is coordinated or to gain any insight into the physical processes affecting the regioselectivity in reactions with electrophiles.

Hence, we embarked on computational studies to understand, on a fundamental atomic and molecular level, processes leading to regio-selectivity in general. To achieve this, we made use of the recently proposed Molecular-Wide and Electron Density-Based (MOWED) approach [29,30]. One must state that the classical approach, i.e., comparative analyses of electronic and Gibbs free energies computed for CIPs, failed as (i) at least four coordination sites were identified, in contradiction with experimental results, and (ii) the classical approach did not provide the information needed to explain regioselectivity on atomic or a molecular fragment level.

In general, in agreement with experimental data, MOWED-based computational modeling showed that coordination of counter ions, Na^+^ and K^+^, to Ade^−^ does indeed take place. Importantly, however, we discovered that the inter-ionic interaction energies differed substantially enough to pinpoint the N3N9-CIP (a bidentate complex) as the most preferred coordination site where the counter ions interact −30 kcal mol^−1^ stronger than in the N7-, N1-CIP, and all out-of-plane CIPs. Considering the Na-Ade complexes, the inter-ionic interaction energy of the bidentate complex was also stronger by about −10 and −12 kcal mol^−1^ relative to the unidentate N9 and N3 coordination sites. These findings can be used to explain the site of alkylation in a benzylation reaction with benzyl bromide under basic conditions in dry DMSO solution where the reactant must be attracted the most to the N3-N9 ‘bottom’ part of Ade^−^, promoting the formation of an N9 benzylation product.

The formation of the CIPs with Na^+^ and K^+^ counter ions showed a strong ion-induced polarization of the adeninate anion. It is about twice as strong for the Na-Ade than K-Ade complexes. The ion-induced charge polarization of the entire Ade^−^ points at the ‘bottom’ part of Ade^−^ (containing N3- and N9-atoms) as most polarizable with these two N-atoms gaining the most negative charge. The ion-induced polarization, in turn, is likely to have a profound impact on chemical reactivity and the preferred site of bond formation with an oncoming electrophile. The adeninate anion of the dominant N3N9-CIP has greater electron density on the N3 and N9 atoms so that these atoms are predicted to be more nucleophilic, which would explain the observed formation of the N9- and N3-alkylated adenine derivatives. On this basis, the N3N9-coordinated CIP should be a starting point in considering the regioselectivity of the adeninate anion towards alkylation. To gain a full picture and explore the regioselectivity specific for a selected electrophile, we recommend MOWED-based modeling of all potential substitution sites. The substitution site with significantly stronger inter-molecular interaction energy would be the most likely and dominant candidate for the new N–C bond formation, or otherwise, a mixture of products should be expected.

We established that the addition of explicit DMSO solvent molecules did not change the trend in the most nucleophilic N-atoms of Ade^−^ for the in-plain M-Ade complexes in M-Ade-(DMSO)_4_ molecular systems. The trend in the most reactive sites, N3,N9 >> N9 > N3 >> N7 >N1, established from the MOWED-based studies in the implicit solvation model holds for the explicit solvation model with four DMSO solvent molecules when the M-Ade-(DMSO)_4_ systems were investigated. Moreover, the study of the strength and nature of interactions between DMSO solvent molecules and counter ions and Ade^−^ revealed the interesting and unexpected chemistry of inter-molecular chemical bonding.

Overall, this study provided evidence in support of the proposed ion pairing of the adeninate anion with the counter ion. It is likely that other nucleobase anions or organic anions generated by deprotonation with a base would similarly form CIPs. This assertion could be investigated using the approach described above. It is clear that the anion experiences electron delocalization upon metal ion coordination, inducing new reactive sites in the ion pair. Without considering these ion-pair states, wrong predictions can be made and experimental results may not be explained using computational analysis. Chemists appropriately consider the simplest model that can explain or predict experimental observations. In the case of the reactions of anionic nucleophiles, the role of the counter ion is usually ignored. This study highlights the importance of considering the metal counter ion in reactions where the anion can react at multiple sites.

## Data Availability

The data presented in this study are openly available in FigShare at https://figshare.com/s/59476a600bad4ba82fc4.

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
