# Peer review of "Coordination Sites for Sodium and Potassium Ions in Nucleophilic Adeninate Contact ion-Pairs: A Molecular-Wide and Electron Density-Based (MOWED) Perspective"

_molecules, 2022, doi:10.3390/molecules27186111_

Round 1
Reviewer 1 Report
The manuscript describes computational studies on the preferred coordination sites of sodium and potassium cations forming ion-pairs with adeninate anions. The study reveals an interesting solvent effect of the DMSO solvent, which directs the coordination of the counter cations into specific nitrogen sites of the anion, therefore increasing the selectivity of the alkylation reactions. Such information is experimentally difficult to obtain, and hence the authors have selected computational approach for the studies. To account for the solvent effects, both explicit and implicit models have been adopted. The selection of the computational methods and models are sound, and the calculations carefully conducted. The nature of the important interactions has been studied via IQA approach. The manuscript is well-written, although at parts a bit long, so some focusing might be in order. Otherwise, the manuscript can be published with minor corrections.
Did the authors consider BSSE when calculating the formation energies via equation (1)? This might have considerable impact on the rather weak energies. However, the authors are probably right in assuming that the direct calculation of the interaction energies would not give definite answers on the preferred coordination sites.
The layout of Figure 1 has been mixed up in the pdf version, so it is very hard to read. Please correct.
Author Response
Reviewer 1
The manuscript describes computational studies on the preferred coordination sites of sodium and potassium cations forming ion-pairs with adeninate anions. The study reveals an interesting solvent effect of the DMSO solvent, which directs the coordination of the counter cations into specific nitrogen sites of the anion, therefore increasing the selectivity of the alkylation reactions. Such information is experimentally difficult to obtain, and hence the authors have selected computational approach for the studies. To account for the solvent effects, both explicit and implicit models have been adopted. The selection of the computational methods and models are sound, and the calculations carefully conducted. The nature of the important interactions has been studied via IQA approach. The manuscript is well-written, although at parts a bit long, so some focusing might be in order. Otherwise, the manuscript can be published with minor corrections.
Did the authors consider BSSE when calculating the formation energies via equation (1)? This might have considerable impact on the rather weak energies. However, the authors are probably right in assuming that the direct calculation of the interaction energies would not give definite answers on the preferred coordination sites.
The layout of Figure 1 has been mixed up in the pdf version, so it is very hard to read. Please correct.
Our response to Reviewer 1:
Thanks you very much for an objective and constructive response and evaluation of our contribution.
Specific answers:
Reviewer 1:
Did the authors consider BSSE when calculating the formation energies via equation (1)? This might have considerable impact on the rather weak energies. However, the authors are probably right in assuming that the direct calculation of the interaction energies would not give definite answers on the preferred coordination sites.
Authors:
Most of our computational studies was performed at the CCSD level and this does not require the BSSE correction. In parallel, we have performed modelling at the B3LYP level and included most of our results obtained in the Supporting Information. The B3LYP data follows exactly the trends obtained at the CCSD level. Hence, we did not implemented the BSSE correction for molecular energies obtained at the B3LYP level.
The reviewer is correct: the direct calculation of interaction energies has not provided definite answers on the preferred coordination site and this is why we embarked on the MOWED approach reported recently.
We are sorry that Figure 1 and 2 have been mixed up in the pdf version sent to the Reviewer. We asked the editorial office and they assured us that they have perfect Figures to be reproduced in the published article and instructed us to do nothing in this regard.

Reviewer 2 Report
The article from Buyens and coworkers deals with the coordination chemistry of Adeninate contact ion-pairs. The article is well written and the evidences support the initial hypothesis made by the authors. Also, the included literature is appropiate and up to date. I just have some concerns prior to publication:
- Figures 1 and 2 are not displayed correctly, since some of the images are cut or overlap with others. The authors should redo both figures.
- Just to complete the picture, I suggest to include a MEP of both the naked adeninate anion as well as of the Ade-Na and Ade-K complexes to evaluate the changes of the electrostatic potential over the different N atoms and the pi system of the adenine ring. This would likely complement and enrich the discussion in section 3.1.1.
- How does REP-FAMSEC compares in terms of accuracy and versatility to other energy partition schemes (e.g. SAPT)?
Author Response
Reviewer 2
The article from Buyens and coworkers deals with the coordination chemistry of Adeninate contact ion-pairs. The article is well written and the evidences support the initial hypothesis made by the authors. Also, the included literature is appropiate and up to date. I just have some concerns prior to publication:
- Figures 1 and 2 are not displayed correctly, since some of the images are cut or overlap with others. The authors should redo both figures.
- Just to complete the picture, I suggest to include a MEP of both the naked adeninate anion as well as of the Ade-Na and Ade-K complexes to evaluate the changes of the electrostatic potential over the different N atoms and the pi system of the adenine ring. This would likely complement and enrich the discussion in section 3.1.1.
- How does REP-FAMSEC compares in terms of accuracy and versatility to other energy partition schemes (e.g. SAPT)?
Our response to Reviewer 2:
Thanks you very much for an objective and constructive response and evaluation of our contribution. We will address your concerns below.
Specific answers:
Reviewer 2:
- Figures 1 and 2 are not displayed correctly, since some of the images are cut or overlap with others. The authors should redo both figures.
Authors:
We are sorry that Figure 1 and 2 have been mixed up in the pdf version sent to the Reviewer. We asked the editorial office and they assured us that they have perfect Figures to be reproduced in the published article and instructed us to do nothing in this regard.
Reviewer 2:
- Just to complete the picture, I suggest to include a MEP of both the naked adeninate anion as well as of the Ade-Na and Ade-K complexes to evaluate the changes of the electrostatic potential over the different N atoms and the pi system of the adenine ring. This would likely complement and enrich the discussion in section 3.1.1.
Authors:
We do appreciate your suggestion very much. Because of the focus of our manuscript on ‘A Molecular-Wide and Electron Density-Based (MOWED) Perspective’, we decided not to follow this suggestion. Any classical and localised or focused on a specific N-atom approach failed to produce data in accord with experimental results. Only when the MOWED approach was implemented we were able to support the experimental data fully and convincingly. Moreover, we want our contribution to be suitable for a general and broad chemist community and hence we tried to minimise technical computational details and very technical and theoretical discussions.
It would be very interesting, however, to compare the predictions made from several classical approaches and recently published MOWED approach. We hope that some researchers will embark on such project that is out of the scope of the present manuscript.
Reviewer 2:
- How does REP-FAMSEC compares in terms of accuracy and versatility to other energy partition schemes (e.g. SAPT)?
Authors:
One must compare ‘apples with apples’. Whereas SAPT and many other EDA schemes are based on orbitals, the REP-FAMSEC energy-partitioning scheme takes an advantage of real space and orbital-independent IQA energy partitioning scheme. There are too many fundamental differences between SAPT and REP-FAMSEC approaches to discuss them in detail. Please read the excellent review by Peifeng Su et al., Generalized Kohn-Sham energy decomposition analysis and its applications, in WIREs Comput Mol Sci. 2020, 10, e1460,
We would like to include in our response the following citation from the above review:
‘Perturbation EDA methods employ the perturbation theory to study intermolecular interactions. The symmetry adapted perturbation theory (SAPT) method developed by Jeziorski et al is the most widely used perturbation EDA method.5,8,9 SAPT is a perturbation theory with double perturbation operators, that is, intra-monomer correlation operators and intermolecular interaction operator. The basic components in SAPT include electrostatic, exchange, induction, and dispersion terms. The advantage of SAPT is its ability to rigidly quantify dispersion. This method is restricted to noncovalent interactions since it is based on the assumption that the magnitude of intermolecular interactions is small. SAPT has been combined with density functional theory (DFT), yielding SAPT(DFT)42,43 and DFT-SAPT.44–46 In SAPT(DFT)/DFT-SAPT methods, the molecular orbitals in Hartree-Fock method are replaced by Kohn-Sham (KS) orbitals to obtain individual interaction terms. SAPT has evolved into many variants, including intra-SAPT for intramolecular interaction,47,48 XSAPT that combines pairwise-additive SAPT with a monomer-based self-consistent field calculation for many-body systems49,50 and so on. In real-space EDA methods, the energy components are built from reduced density matrices. It partitions the physical space into physically meaningful domains associated with the interacting molecules. The interacting quantum atoms (IQA) method is one of the representative real-space EDA methods.51–53 It considers domains bound by the zero-flux surfaces of electron densities as defined in quantum theory of atoms in molecules (QTAIM). QTAIM divides the real space into atomic basins where the interatomic surfaces are zero-flux surfaces of the gradient of electron densities.54 After the space is partitioned into QTAIM basins, the IQA method simply proceeds by partitioning all the mono- and bielectronic integrals into intra- and inter-fragment energetic components respectively.’